# PCSK9: A Multi-Faceted Protein That Is Involved in Cardiovascular Biology

**DOI:** 10.3390/biomedicines9070793

**Published:** 2021-07-08

**Authors:** Sai Sahana Sundararaman, Yvonne Döring, Emiel P. C. van der Vorst

**Affiliations:** 1Interdisciplinary Centre for Clinical Research (IZKF), RWTH Aachen University, 52074 Aachen, Germany; ssahanasunda@ukaachen.de; 2Institute for Molecular Cardiovascular Research (IMCAR), RWTH Aachen University, 52074 Aachen, Germany; 3Department of Pathology, Cardiovascular Research Institute Maastricht (CARIM), Maastricht University Medical Centre, 6229 ER Maastricht, The Netherlands; 4Institute for Cardiovascular Prevention (IPEK), Ludwig-Maximilians-University Munich, 80336 Munich, Germany; 5German Centre for Cardiovascular Research (DZHK), Partner Site Munich Heart Alliance, 80336 Munich, Germany; 6Department of Angiology, Swiss Cardiovascular Center, Inselspital, Bern University Hospital, University of Bern, 3010 Bern, Switzerland

**Keywords:** PCSK9, cardiovascular disorders, low density lipoprotein receptor, cholesterol, polymorphisms, monoclonal antibodies

## Abstract

Pro-protein convertase subtilisin/kexin type 9 (PCSK9) is secreted mostly by hepatocytes and to a lesser extent by the intestine, pancreas, kidney, adipose tissue, and vascular cells. PCSK9 has been known to interact with the low-density lipoprotein receptor (LDLR) and chaperones the receptor to its degradation. In this manner, targeting PCSK9 is a novel attractive approach to reduce hyperlipidaemia and the risk for cardiovascular diseases. Recently, it has been recognised that the effects of PCSK9 in relation to cardiovascular complications are not only LDLR related, but that various LDLR-independent pathways and processes are also influenced. In this review, the various LDLR dependent and especially independent effects of PCSK9 on the cardiovascular system are discussed, followed by an overview of related PCSK9-polymorphisms and currently available and future therapeutic approaches to manipulate PCSK9 expression.

## 1. Introduction

Pro-protein convertase subtilisin/kexin type 9 (PCSK9) is a soluble protease that has been widely studied in the field of cholesterol homeostasis and in cardiovascular biology after its discovery in 2003 [1]. Originally named Neural Apoptosis Regulated Convertase 1 (NARC1), PCSK9 is part of the secretory serine proteinase family called Pro-protein Convertases (PCs) [2]. Hepatocytes are the primary source of PCSK9 that is secreted into the circulation. However, also other cells can produce and secrete PCSK9, like cells in the intestines [3,4], pancreas [5], adipose tissue [6], kidneys [7] and brain [8]. Interestingly, the circulating levels of PCSK9 biologically increases in the late night, and decreases in the late afternoon, following a diurnal rhythm [9]. Furthermore, the total circulating PCSK9 levels are influenced by sex as females have higher levels compared to men, suggesting that hormones like oestrogens are involved in the expression and secretion of PCSK9 [10,11,12]. Besides sex, also age, body mass index (BMI), plasma cholesterol and triglyceride levels and blood pressure have been shown to modulate the total concentration of PCSK9 [10]. The best-known function of PCSK9 is its effect on low density lipoprotein receptor (LDLR), to which it can bind and thereby facilitate its lysosomal degradation. Lately, PCSK9 is the focus of various studies in different clinical contexts as a marker for cardiovascular risks [13,14,15,16,17] independent of other known traditional risk factors and independent of its influence on LDLR. PCSK9 is thought to play an important role in cardiovascular diseases (CVDs) via different mechanisms, either through binding to Epidermal Growth Factor (EGF) domains on receptors or through binding with receptors present in lipid rafts and associated cell membrane micro-domains or by influencing the gene expression and protein response of various factors involved in the development of cardiac complications [18]. Further supporting its role in CVDs, is the fact that various gain-of-function (GOF) mutations of PCSK9 are associated with hypercholesterolaemia and thereby an elevated risk for cardiac events [19,20,21,22,23]. In addition to these GOF mutations, also hundreds of loss-of-function (LOF) mutations of PCSK9 have already been discovered, that hinder its secretion into circulation providing protection against cardiovascular complications [24,25,26,27]. Targeting PCSK9 using monoclonal antibodies (mAbs) has emerged as an additional and additive therapy to treat hyperlipidaemia due to its regulation of LDLR [28,29,30,31]. However, several recent studies highlighted that PCSK9 can also have various additional effects that are independent of the LDLR, that provide more information on the involvement of PCSK9 in cancer, type 2 diabetes, obesity and several cardiovascular disorders [32,33,34].

Therefore, in this review the biological and physiological characterisation of PCSK9 will first be outlined in the context of cardiovascular biology, followed by an overview of the LDLR dependent and especially the LDLR independent effects of PCSK9 in this context. Furthermore, we will focus on state-of-the-art PCSK9 inhibitory techniques and their clinical potential.

## 2. PCSK9 Biology

The 22kb human PCSK9 gene is located on the chromosome 1p32 [2]. It has 12 exons and 11 introns that encode a 692 amino acid proteinase [2]. The protease is manufactured within the endoplasmic reticulum (ER) and has a molecular mass of 120 kDa [2]. It is secreted as an inactive protein that later undergoes post-translational modifications to form a 62 kDa mature protein. PreProPCSK9 consists of five segments: a signal peptide (amino acids 1–30), a N-terminal prodomain (amino acids 31–152), a catalytic domain containing the active sites (amino acids 153–404), a C-terminal cys-his-rich domain (amino acids 455–692) and a small peptide that serves as a link between the catalytic and C-terminal domain also known as the hinge region (amino acids 405–454) [35]. The C-terminal domain can further be divided into three modules, namely M1, M2 and M3. It has been demonstrated that the M2 module plays a crucial role in the extracellular PCSK9-LDLR binding [36]. The maturation or the formation of active PCSK9 requires three steps (Figure 1): first, the signal sequences are cleaved; pro-PCSK9 then undergoes proteolysis; the last step is the transportation of the protein into the circulation via the Golgi complex [37]. PCSK9 is secreted into the circulation as an enzymatically active protein. The secreted PCSK9 has only three domains and is a heterodimer protein [38]. The first is the pro-domain that is cleaved auto-catalytically, nevertheless it remains attached to the mature protein non-covalently to act as an escort for its intracellular movement. The pro-domain region contains acidic residues that have an auto-inhibitory role by interacting with the basic residues present in the catalytic domain [39]. The catalytic domain contains the proteolytic active site, which is necessary for the autocatalytic cleavage, but is not related to LDLR-PCSK9 binding [38]. Lastly, the C-terminal domain consisting of the three modules is attached to the catalytic domain [38]. While PCSK9 is a serine protease, the degradation of LDLR does not require proteolytic activity of PCSK9, only the ability of PCSK9 to chaperone LDLR towards lysosomes [40].

PCSK9 also exists in a furin-cleaved form that is 55 kDa in size [41] (Figure 1). This form is slightly less active and has half the affinity to LDLR as well as a shorter half-life compared to the intact heterodimer form. In addition, the furin-cleaved form is in general not associated with ApoB containing lipoproteins [41], establishing that the furin-cleaved form is the less active one. PCSK9 is cleaved by furin at the N-terminal to exclude the pro-domain and releases the protein into circulation [42]. This cleavage can occur only in circulation [42], because there is no evidence that furin can cleave PCSK9 intracellularly although they co-exist in the Golgi complex. The fact that the furin-cleaved form does not contain the pro-domain to act as a chaperone for PCSK9 to get secreted extracellularly validates the hypothesis that furin only cleaves PCSK9 in circulation [42].

Various transcription factors and cofactors regulate the expression of the PCSK9 gene. The PCSK9 promoter contains an important region that is essential for transcription, called the Sterol regulatory Element (SRE) [43]. Sterol-response Element binding protein (SREBP-1/2) is a predominant transcription factor that connects to this SRE promoter in PCSK9 [35]. Low dietary cholesterol concentrations upregulate SREBP-1/2 expression that in turn regulates PCSK9 levels in circulation [44]. Additionally, hepatocyte nuclear factor 1 α (HNF1α) and forkhead box O3 (FoxO3) are also predominantly involved in the transcription of PCSK9 [43]. HNF1α acts by binding to a site that is located next to the SRE site in the promoter region [43], while FoxO3 suppresses the transcriptional activity of HNF1α by recruiting sirtuins 6 instead of HNF1α [45]. Furthermore, sirtuins 1 and 6 and histone deacetylases are known to supress SREBP2 expression in the liver resulting in suppression of PCSK9 expression and thereby decreasing the circulating PCSK9 levels [44]. Insulin is also involved in the regulation of PCSK9 expression as it could be shown that excess insulin in the blood decreases the transcription of the PCSK9 gene [46] in post-menopausal obese women, but it does not amend the mRNA levels in healthy men and type-2 diabetic patients [47]. Peroxisome proliferator-activated receptor α and γ (PPARα and PPARγ) regulate the gene expression of PCSK9 as well [2], PPARγ increases the gene expression while PPARα decreases it. Furthermore, PCSK9 interacts with sortilin in the Trans-Golgi complex, which is a transmembrane protein involved in the development of atherosclerosis and other CVDs [48], aiding in the extracellular secretion of PCSK9. It could be shown that mice that overexpress sortilin have increased circulating PCSK9 levels, while sortilin deficient mice have low circulating PCSK9 levels, suggesting that this protein–protein interaction is highly essential for the regulation of secretion of PCSK9 by hepatocytes [49]. The binding of PCSK9 to sortilin occurs at the pH 6.5 and any modification of the pH deteriorates the binding [49].

Although a lot of progress has been made over the last years to fully understand the regulation of PCSK9, it is still not completely understood as this regulation turns out to be highly dynamic. The following sections will discuss how PCSK9 not just interacts with LDLR, but also with several other receptors to influence cardiovascular biology.

## 3. PCSK9-LDLR Binding

In 2007, PCSK9 emerged as a novel target to treat dyslipidaemia. Numerous studies have identified a clear link between low density lipoprotein cholesterol (LDL-C) and PCSK9. Normally, LDL-C binds and gets internalised by binding to the LDLR, thereby mediating its clearance by endocytosis. In the presence of PCSK9, the LDLR undergoes lysosomal degradation, leading to an inhibition of the recycling of the receptor and therefore an increase in the plasma levels of LDL-C [50]. This degradation of the LDLR by PCSK9 occurs through two different pathways, an extracellular and intracellular one (Figure 2). The commonly known and widely studied process of LDLR degradation is by extracellular binding of PCSK9 to the cell surface of the LDLR. PCSK9 does not degrade LDLR itself, it just acts as a tag to promote the degradation process. The catalytic domain of the mature circulating PCSK9 binds to the epidermal growth factor A (EGF-A) domain of the LDLR. The PCSK9-LDLR compound then undergoes endocytosis through clathrin-coated pits and is taken up by the endosomes or lysosomes in the cells, leading to the deprivation of LDLR as well as PCSK9 [2,50]. This PCSK9-LDLR binding is dependent on calcium concentrations and is affected by pH changes as acidic pH of the endosome increases the affinity of PCSK9 to LDLR [51]. This increased affinity is caused by an interaction between the ligand-binding domain of the LDLR and the C-terminal domain of the mature PCSK9 [50]. This binding complex occurs in the endosome and the binding sustains, causing LDLR to not unfold and therefore not getting recycled back to the cell surface. The C-terminal domain of PCSK9 is quite important for its binding to LDLR, but alone it has no influence on the LDLR [39]. The interaction between the catalytic domain of PCSK9 and the EGF-A domain of LDLR contributes less to the LDLR degradation, while the positive charge at acidic pH of C-terminal domain and its size is rather responsible for the PCSK9-LDLR binding [52]. Although a review published in 2017 concluded that only hepatic LDLR is affected by circulating PCSK9 [53], studies have revealed that PCSK9 is involved in regulating the expression of LDLR in pancreas and other organs as well [5]. PCSK9 also facilitates intracellular degradation of mature LDLR present in the Golgi or Trans-Golgi complex, before it reaches the cell surface [44]. In this pathway, the immature or the budding PCSK9 binds to the LDLR found in the Golgi apparatus and redirects it to the lysosomes for degradation. Unlike the extracellular degradation of LDLR, this intracellular degradation compels the involvement of catalytic activity of PCSK9 [44].

On the other hand, PCSK9 also interacts with LDLR in non-destructive ways (Figure 2). The catalytic domain of the pre-pro-PCSK9 binds to the EGF-A repeat of the precursor of LDLR in the ER that is essential for the transport of LDLR to the Golgi apparatus wherein LDLR matures by obtaining various carbohydrates [44,54]. This binding is not just advantageous for the LDLR, but it also helps pre-pro-PCSK9 to undergo auto-catalytic cleavage and thus undergo maturation. The mature PCSK9 then tethers with an additional salt bridge to LDLR and this complex is released by the cell to increase the number of cell-surface receptors [54]. Nonetheless, this chaperone activity is not critical for the transportation of LDLR to cell surfaces and happens much less often than the degrading activity, considering that even in the absence of PCSK9 the LDLR is transported to the cell membrane without any hindrance [54]. Although, the reason why PCSK9 is able to chaperone LDLR to the surface without leading to lysosomal degradation is not known yet.

The binding of PCSK9 to LDLR resulting in its degradation plays an important role in the cholesterol homeostasis, as it could be shown that a full-body knockout of PCSK9 in mice leads to an extreme reduction of cholesterol levels in the serum. More particularly, liver-specific PCSK9 knockout also leads to a considerable reduction in circulating cholesterol levels concluding that the hepatic PCSK9 contributes most to the cholesterol phenotype [2]. The destruction of LDLR by PCSK9 leads to hyperlipidaemia, which is associated with numerous cardiovascular complications. In particular, lipid uptake and accumulation by macrophages in the vessel wall and formation of ‘foamy’ macrophages leads to the development of atheromas. PCSK9 inhibition using mAbs reduces the accumulation of lipid particles inside monocytes and thereby also inhibits monocyte chemotaxis [55]. Apart from the effects on lipid uptake, silencing of PCSK9 using siRNAs also increases the expression of the chemokine receptor CCR2 on monocytes, thereby increasing their potential to migrate towards and infiltrate the arterial wall, triggering arterial inflammation and promoting atherogenesis [56]. The degradation of hepatic LDLR by PCSK9 also causes hypertriglyceridaemia, due to increased amounts of circulating ApoB containing lipoproteins that could not be degraded by LDLR [53]. This has been confirmed by several murine models where overexpression of PCSK9 leads to an increase in the plasma ApoB particles and VLDL-triglycerides, whereas deficiency of PCSK9 results in a decline of triglyceride levels in the plasma of these mice [53]. Therefore, the degradation of LDLR influences not just the cholesterol homeostasis, but also the triglyceride homeostasis, that eventually leads to diversified pathologies in the vasculature and beyond.

## 4. PCSK9′s Activity Independent of LDLR

The function of PCSK9 is not just limited to the regulation of the LDLR, it also influences the lysosomal degradation of various receptors that structurally relate to the LDLR. Receptors such as very low density lipoprotein receptor (VLDLR), LDLR-related protein 1 (LRP1) [53] and apolipoprotein E receptor 2 (ApoER2) [57] also contain EGF-A domains enabling the interaction with PCSK9. The expression of PCSK9 by vascular endothelial cells (ECs), smooth muscle cells (SMCs), cardiomyocytes along with various immune cells also sparks our interest to substantially acknowledge the role of PCSK9 in pathologies of the cardiovascular system. In this section, we review the myriad of recently published studies that relate the LDLR-independent effects of PCSK9 on vascular diseases that include atherosclerosis, myocardial infarction and calcification and discuss the assorted mechanisms by which PCSK9 targets the cardiovascular system (summarised in Figure 3).

### 4.1. Inflammation and Atherosclerosis

Atherosclerosis is characterised by chronic inflammation of the vasculature, accompanied by lipid retention in arteries and the formation of plaques. Activation of the endothelial cell layer by atherogenic and pro-inflammatory stimuli initiates atherogenesis, which is followed by the infiltration of monocytes, lymphocytes, and various other immune cells [58]. The recruited monocytes differentiate into macrophages and internalise atherogenic lipoproteins and transform into foam cells. This lipoprotein and thus cholesterol accumulation continues which will eventually result in apoptosis and secondary necrosis, forming a lipid-rich and necrotic core in the plaques. Meanwhile, during disease progression medial SMCs proliferate and migrate to form a fibrous cap that stabilises the plaques and shields the necrotic core [2]. If this cap is broken down in later stages of the disease, plaques can rupture leading to thrombus formation, one of the primary causes of cardiovascular complications and related deaths. Albeit the steps involved in atherogenesis have already been clearly outlined by various studies, the exact role of PCSK9 in atherosclerosis has only recently been (partly) unravelled.

PCSK9 is expressed on aortic ECs, SMCs, macrophages, dendritic cells and epithelial cells [57], suggesting that PCSK9 not only regulates atherosclerosis by influencing serum LDL-C levels, but also by interacting and influencing cellular processes in the vessel wall to aggravate atherosclerosis [33,34]. The severity of atherosclerosis is positively correlated with circulating PCSK9 levels [14]. Furthermore, it could be demonstrated that low shear stress, such as in the aortic arch branch points and aorta-iliac bifurcations, increases the expression of PCSK9 by ECs, while high shear stress does the opposite [59,60,61]. Studies observed that silencing of PCSK9 reduces the levels of inflammatory cytokines in aortic tissues, resulting in an attenuated plaque inflammation without affecting LDLR levels [62,63]. This effect is caused by the fact that PCSK9 influences the inflammatory pathways to activate NF-κB and upregulates Toll-like receptor 4 (TLR4) expression [62]. The same study also identified that PCSK9 is overexpressed in atherosclerotic plaques of *Apoe^−/−^* mice [62]. Further confirming an important role of PCSK9 in inflammation and apoptosis, it could be shown that silencing of PCSK9 in ECs in vitro reduces the ability of the cells to go into apoptosis when exposed to oxLDL by reducing pro-apoptotic factors Bcl2-associated protein (Bax), Caspase 3 and 9, while increasing the anti-apoptotic factor Bcl-2 as well as activating p38/JNK/MAPK pathways [2]. Additionally, PCSK9 induces pyroptosis, mitochondrial dysfunction and reactive oxygen species (ROS) production in human umbilical vein endothelial cells (HUVECs) after an exposure to oxLDL, suggesting that PCSK9 also plays a valuable role in the antioxidant response in the context of atherosclerosis [64]. The increased expression of PCSK9 by low shear stress also induces ROS generation via the nicotinamide adenine dinucleotide phosphate (NADPH) oxidase system [60], clearly demonstrating an important role of PCSK9 in ECs.

Besides ECs, vascular SMCs are also affected by shear stress when the EC layer is disrupted, as it could be demonstrated that low shear stress upregulates their proliferation and migration capability while increasing the secretion of PCSK9 by the SMCs. Several studies have demonstrated that SMCs secrete functional PCSK9 into the atheroma that exerts effects on monocytes migration in the intima. The overexpression of PCSK9 by SMCs in atherosclerotic plaques also reduces the ability of macrophages to ingest aggregated LDL (agLDL) and oxLDL molecules through scavenger receptors and LDLR related proteins [4,65]. PCSK9 secreted by SMCs not just plays a paracrine effect, PCSK9 also regulates the metabolism in SMCs. This could be perceived by several studies: for instance, treating SMCs in vitro with recombinant PCSK9 stimulates mitochondrial damage that in turn activates apoptosis pathways [66]. Studies were performed in vitro to validate this, and it was seen that mice that are deficient in PCSK9 show less mitochondrial damage in SMCs compared to wild type mice when injected with LPS [66]. Mediated by mitochondrial ROS generation, PCSK9 and mitochondrial DNA damage influence each other in a positive feedback loop to facilitate cell injury and thereby advance atherosclerosis [56]. Contrarily, PCSK9 might provide a protective effect against atherosclerosis progression by regulating SMCs. Deficiency of PCSK9 in mice has been shown to reduce the ability of the SMCs to proliferate and migrate, with the cells expressing more than usual levels of contractile, such as alpha-actin and myosin proteins [2,67]. These SMCs also express very low levels of synthetic proteins, such as extracellular matrix components and collagen that are involved in the formation of fibrous cap [68]. Combined, the lack of PCSK9 therefore seems to reduce the fibrous cap formation and thereby destabilises the lesions. Altogether, it could be shown that SMCs do not only express PCSK9, but that PCSK9 can also influence cellular processes in SMCs to influence plaque stability.

Besides its influence on vascular cells, PCSK9 has also been shown to exert pro-inflammatory and pro-atherogenic effects on macrophages in vitro even in the absence of LDLR [2]. For example, PCSK9 has been shown to inhibit ATP-binding cassette transporter (ABCA1) mediated cholesterol efflux in macrophages and thereby disturbs the cholesterol homeostasis [69]. Furthermore, PCSK9 increases the infiltration of Ly6c^hi^ monocytes into the atherosclerotic plaques [70]. Inhibition of PCSK9 also supresses the expression of inflammatory cytokines IL-1α, IL-6, IL-1β, MCP-1 and TNFα and the activation of NF-κB pathway when macrophages are exposed to oxLDL and inflammation [56,62]. In line with this, macrophages that are stimulated with recombinant PCSK9 express pro-inflammatory cytokines in a dose-dependent fashion [71]. These pro-inflammatory effects are LDLR-independent as it could be shown that PCSK9 has similar effects on macrophages from LDLR^−/−^ mice [72]. Macrophages can also secrete PCSK9 themselves and in vitro and in vitro experiments have discovered that the NLR family pyrin domain containing 3 (NLRP3) inflammasome triggers the expression of PCSK9 in macrophages by IL-1β release [73]. PCSK9 secreted from lipid-loaded macrophages can also regulate several LDLR like receptors. For example, members of the LRP family, such as LRP5 form a complex with PCSK9, that is highly essential for further lipid uptake by macrophages. When this LRP5-PCSK9 bond is formed, the complex triggers the TLR4/NF-κB pathway resulting in increased inflammation [74]. Similarly, PCSK9 upregulates the expression of various scavenger receptors such as SR-A, CD36 and LOX-1 to boost the uptake of oxLDL by macrophages, to further facilitate inflammation [56,75]. There is also a feed-forward loop in which the activation of LOX-1 triggers PCSK9 expression as well [67]. Although this process can play an import role in atherogenesis, the exact effects of inhibition of PCSK9 on LOX-1 remain elusive [53].

Considering all the above-mentioned studies, it can be summarised that PCSK9 has a direct effect on both vascular and immune cells to directly influence atherogenesis not only by modulating the LDLR but also by LDLR-independent mechanisms.

### 4.2. Myocardial Infarction

Complete cessation or lowering of blood flow to parts of the heart, due to atherosclerosis formation, causes the myocardium to be deprived of oxygen, resulting in a myocardial infarction (MI) which is associated with myocardial cell death and necrosis [76]. PCSK9 levels were observed to be upregulated in rats after acute MI [77], which is also validated in humans as serum levels of PCSK9 are also elevated in patients with acute MI [11,60]. This could be explained by the fact that acute MI leads to an increase in the expression of SREBP-2, hepatocyte nuclear factor 1 α (HNF1α) and NLRP3 resulting in an elevated expression of PCSK9 [78]. Furthermore, MI leads to hypoxia of cardiomyocytes and it was demonstrated that hypoxia induces the expression of PCSK9 in cultured cardiomyocytes [60,79]. The PCSK9 produced by hypoxic cardiomyocytes promotes injury even in healthy cardiomyocytes [79]. Moreover, the elevated PCSK9 levels also stimulate the secretion of pro-inflammatory cytokines and activate NF-κB signalling in the recruited macrophages at the sites of injury [79]. This can be further confirmed by inhibiting or knocking out PCSK9 in murine models as PCSK9 deficiency improves the cardiac function and reduces the infarct size in mice [80]. Consistent with this, when mice are subjected to coronary artery occlusion, the mice develop infarcts and the area surrounding the infarct zones have more than normal, non-infarcted, levels of PCSK9 protein expression [60]. These zones with elevated PCSK9 expression also show elevated autophagy. Inhibiting PCSK9 reduces the size of the infarct and rescues the phenotype, suggesting that PCSK9 plays a causal role in the development of these infarcts.

MI is very closely associated with coronary artery disease (CAD), as MI causes occlusion of coronary arteries [81]. A study observed that PCSK9 levels in patients with a history of CAD are associated with high levels of circulating cholesterol, triglyceride, and inflammation which did not correspond to the severity and progression of the disease [82], although contradicting studies also exist [83]. PCSK9 is involved in the regulation of diverse lipoproteins involved in the development and progression of CAD. For example, oxLDL induces the expression of PCSK9 in cardiomyocytes which, in turn, reduces the cardiomyocyte cell shortening [84]. Serum lipoprotein (a) (Lp(a)) is yet another high-risk factor for CADs, but the relation between Lp(a) and PCSK9 remains debateable. Different studies revealed a positive association between serum Lp(a) and PCSK9 [85], however, other studies did not find associations between these two [86].

To summarise, in situations of MI, the expression of PCSK9 is elevated to not just alter the lipid metabolism but also to promote the cytotoxic capacity of oxLDL and increase the expression of apoptotic markers and promotes autophagy [78]. Inhibition of PCSK9 is believed to play a role in MI by limiting vascular remodelling and by tampering autophagy and inflammatory markers on top of influencing the lipid profile and cardiac function, although studies are yet to find all the mechanistic insights [87].

### 4.3. Obesity Induced CVD

Obesity being a major health problem at the moment, leads to the development of metabolic comorbidities in the form of cardiovascular complications. Obesity is characterised by the enlargement of adipocytes and adipose tissue, and secretion of inflammatory cytokines to instigate pathological complications to the cardiovascular system [88]. At first, it was believed that adipocytes do not express PCSK9, but only express targets of PCSK9 like VLDLRs and LDLRs on their surface [89]. On the contrary, recent studies have discovered the presence of PCSK9 in human adipocytes, that is easily detectable on gene and protein levels [16] although these levels are relatively low compared to the levels of hepatic PCSK9. Studies have identified that obesity upregulates the expression of PCSK9 and high levels of PCSK9 are again associated with a progression of the disease, therefore making it a vicious feedback cycle between adiposity and PCSK9. The levels of PCSK9 expression in adipose tissue positively correlates with the body mass index (BMI) of the individual, suggesting that obesity and adiposity induces the expression of PCSK9 [16]. Likewise, excessive dietary fat consumption leads to an upregulation of hepatic PCSK9 [90]. The function of PCSK9 in relation to obesity seems controversial. Validating the hypothesis that PCSK9 provides a defence mechanism against obesity, studies showed that visceral adipose tissue content increases in PCSK9 knockout mice due to increased expression of VLDLR [44]. PCSK9 targets VLDLR and ApoER2, receptors that are responsible for the hydrolysis of triglyceride-rich lipoproteins (TRLs). This hydrolysis is essential for fat storage in adipose tissues as well as utilisation of fat by the vascular tissues [41]. Through such VLDLR regulation, PCSK9 limits adipogenesis in visceral adipose tissue and protects against adiposity [89]. On the contrary, PCSK9 also influences the expression of receptors and molecules other than VLDLR to promote obesity. For example, PCSK9 is also involved in the degradation of CD36, reducing fatty acid uptake and cholesterol/triglyceride accumulation in adipose tissue and liver [91]. Furthermore, the PCSK9 secreted from adipocytes is involved in the modification of myocardial LRP1 levels and the glucose metabolism in the cardiomyocytes by influencing C1q/tumour necrosis factor-related protein-9 (CTRP9) [6], which is involved in various cardiac complications [92]. In conclusion, adipose tissue expresses and secretes PCSK9 which via LDLR-independent mechanisms can contribute to obesity and thus CVDs.

### 4.4. Calcification

Ectopic calcification is associated with old age and is a very relevant comorbidity of CVDs. Calcific aortic valve stenosis (CAVS) is a form of ectopic calcification. Studies have indicated that there is an association between the levels of plasma PCSK9 and the severity of aortic valve stenosis [93]. The expression of PCSK9 is higher in the calcified valves of patients suffering from CAVS, and the valvular interstitial cells isolated from these calcified valves show elevated expression of markers of calcification with elevated PCSK9 expression [94]. In line with this, aortic valves isolated from PCSK9^−/−^ mice show lower markers for calcification compared to mice with functional PCSK9 [93]. Histological examination of aortic samples from patients with coronary artery calcification further validated the abnormal expression of PCSK9 in calcified area [95]. As a validation of the causal role of PCSK9 in vascular calcification, LDLR^−/−^ mice overexpressing PCSK9 using an adenovirus containing PCSK9 develop more vascular calcification in response to being fed with Western diet compared to mice with normal PCSK9 levels [96]. All these studies suggest that PCSK9 is a marker and perhaps even a causal factor of vascular calcification that deserves more attention in future research.

## 5. PCSK9 Polymorphisms

Several polymorphisms occur in the PCSK9 gene that tend to cause an effect on CVDs [19,97,98,99,100,101,102,103,104]. For example, the PCSK9 gene undergoes loss-of-function (LOF) mutations that disturb the secretory pathways of PCSK9 into circulation. Certain LOF mutations of PCSK9 result in the hindrance of the transport of PCSK9 from the ER complex to the Golgi apparatus [44,105], leading to reduced circulating levels of mature or furin-cleaved PCSK9. Additionally, nonsense mutations to PCSK9 also give protection against CAD by reducing the LDL-C levels [106]. LOF mutations are also related to an increase in the proteolysis [39,107,108] and lowering of the levels of lp(a), LDL-C and reduce the risk of cardiac complications such as MI and aortic valve stenosis [109,110]. Furthermore, LOF mutations attenuate the cytokine response in healthy as well as septic patients when they are administered LPS [60]. Arg46Leu and Asp301Gly are two LOF mutations, that can be inherited from the parents to the children, and they have been associated with lowering LDL-C levels and providing protection against cardiovascular pathologies [27]. Mutations in the hinge region of the protein can be caused by variants W428X, A443T and R434W, resulting in misfolded and non-functional PCSK9 [111]. In conclusion, LOF mutations result in low circulating LDL-C levels and reduce the risk of developing CVDs [53].

The PCSK9 gene also undergoes gain-of-function mutations (GOF), that usually result in hypercholesterolaemia [112] causing accelerated vascular aging and CVDs [113]. For instance, mutation D374Y enhances the protein to self-assemble to form dimers and trimers [44,114], leading to extreme hypercholesterolaemia and severe atherosclerosis [44]. This mutation is therefore widely used as atherosclerotic models in murine studies. Furthermore, S127R and D374Y mutations increase the affinity of PCSK9 towards LDLR as well as increase the level of every apoB100-containing lipoprotein in the plasma [39,115]. Other single nucleotide polymorphisms (SNPs) increase the intima-medial thickness of the arteries [116], cause arterial plaques [117] and stimulate the progression of CVD [107]. Some variants of PCSK9 show both LOF and GOF. For instance, S127R is a mutation that is involved in both apoB synthesis as well as its catabolism [118]. This variant causes increased production of VLDL and IDL apoB-100 and increases the catabolism of LDL independent of its effect on VLDL and IDL [118]. On the other hand, the variants A245T and R272Q undergo more autocatalytic cleavage than the normal PCSK9, but they do not have any functional consequences on the degradation of LDLR [119]. Additionally, variants I474V and E670G cannot be categorised into LOF or GOF mutations, as functional studies for the mutations are not available [120]. Table 1 and Table 2 summarise the main LOF and GOF mutations that are currently known and provide an overview of the functional consequences of these variants on the pathology of cardiovascular biology.

## 6. PCSK9 Activators/Inhibitors

Cholesterol lowering therapies are the gold standard to reduce the risks of cardiovascular mortality and morbidity. One of the most effective ways to lower circulating cholesterol in blood is the usage of statins. Statins increase the activity of LDLRs, thus increasing the catabolism of VLDL, LDL and intermediate-density lipoprotein (IDL). Furthermore, statins also decrease the hepatic and endogenous cholesterol production by inhibiting HMG-CoA reductase [43], but they do not affect Lp(a) [145]. Interestingly, it could be demonstrated that statins also have some detrimental effects, for example by increasing the levels of circulating PCSK9 [146]. Therefore, inhibiting PCSK9 along with statin therapies has become an important and attractive addition in managing hypercholesterolaemia. In this section we will discuss the numerous studies that have identified various mechanisms to inhibit PCSK9.

Studies have identified numerous naturally occurring inhibitors and small molecule inhibitors to rescue PCSK9 induced CVDs. In the context of naturally occurring substances, berberine has the ability to exert inhibitory effects on the transcription and translation of PCSK9 [147]. Berberine is present in the roots and stems of the plant species *Berberis* and other flowering plants such as *Coptis rhizomes* and *Hydratis Canadensis* [148]. Berberine also increases the mRNA and protein levels of LDLR, independent of influencing PCSK9 [148]. Similarly, a protein-rich grain known as Lupin has been shown to decrease the levels of circulating PCSK9 and inhibits the binding of PCSK9 to LDLR [148]. Furthermore, polyphenols which are present in fruits, vegetables, seeds, herbs, tea, red wine, and nuts [148], are also involved in the inhibition of the PCSK9 protein. For example, a polyphenolic compound called resveratrol downregulates the expression of the SREBP-1c pathway and thereby downregulates the expression of PCSK9 [149]. In contrast, another polyphenol called quercetin activates the transcription of SREBP2 to upregulate the expression of the LDLR gene and reduces the expression of PCSK9 [148]. Furthermore, liraglutide is a glucagon-like peptide-1 (GLP-1) receptor agonist that has been used clinically as anti-diabetic and anti-obesity treatment, which has also been identified as a potent suppressor of PCSK9 expression, explaining at least partly its beneficial effect on CVDs [150]. Other than small molecule inhibitors, RNA aptamers also specifically bind to a target protein with high affinity. One study developed a novel RNA aptamer, called PCSK9-binding RNA (PBR), that binds to PCSK9 with a higher affinity than LDLR and thereby reduces the destruction of LDLR [151].

The administration of monoclonal antibodies (mAbs) against PCSK9 is a novel lipid-lowering therapeutic approach, that inhibits the attachment of PCSK9 with LDLR and LDLR-like receptors [152]. They act on the PCSK9 protein itself rather than targeting the gene expression. Additionally, PCSK9 mAb decreases the production rate of hepatic Lp(a) particles [145] on top of increasing the clearance of them [78]. Thus, the use of mAbs have been highly sought out for the treatment of cardiovascular complications. The inhibitors currently in use to hinder the interaction between PCSK9 and LDLR are alirocumab and evolocumab [1]. Both these inhibitors are used alone or in combination with standard lipid-lowering therapies in adults to reduce the risk of several CVDs such as MI, stroke and atherosclerosis [153], and are administered subcutaneously. Long term administration of mAbs against PCSK9 is safe and results in a reduction of inflammation, arterial wall plaques and the risk of cardiovascular events [154]. Even though the efficacy of mAbs is profound, they require a subcutaneous injection once or twice a month and are therefore still rather expensive. The limitation of frequent administration, high production costs and oral unavailability is one of the major points that should be tackled in the upcoming year to enable the large-scale use of mAbs in the clinic.

As an alternative to mAbs, creating an immune response against PCSK9 through vaccination can be used to provide protection against hyperlipidaemia. In fact, several groups have already developed extremely potent vaccines against PCSK9. For instance, PCSK9Qβ-003 is made of Qβ viral particles to decrease total cholesterol and LDL-C levels in *Apoe^−/−^* mice that eventually reduces the atherosclerotic lesion sizes and makes the plaques more stable [155]. AT04A is another peptide-based vaccine that generates PCSK9 specific antibodies in mice that have the ability to bind to PCSK9 thereby removing it from the circulation [156]. Peptide based vaccines need to be conjugated with carrier proteins in order to cause sufficient immune responses and they fail to promote recognition of B cells. To rectify these disadvantages, a vaccine that can induce large titres of PCSK9 antibody called the PCSK9 multicopy display nano-vaccine (PMCDN) was developed. This nano-vaccine is constructed by self-assembled carrier proteins, passes through lymph nodes into circulation easily and improves endocytosis of PCSK9 in murine models [157]. Additionally, on the surface of nanoliposomes, a peptide to induce high IgG antibodies called the immunogenic fused PCSK9 tetanus has been developed. This vaccine formulation, termed as Nano-liposomal Immunogenic Fused PCSK9-Tetanus with alum vaccine adjuvant (L-IFPTA^+^) was tested in mice with dyslipidaemia, where it inhibits the interaction between PCSK9 and LDLR and reduces the progression of atherosclerosis with long term immunity effects and efficacy [158].

Most small molecule inhibitors, mABs and vaccines target extracellular PCSK9, while there are also ways to inhibit PCSK9 intracellularly via modulation of the gene expression [159]. Initially, antisense oligonucleotides (ASOs) were first used in murine and monkey models to inhibit the translation of the PCSK9 mRNA, but studies were terminated due to unknown reasons [160]. Nevertheless recently, a group has developed a highly potent ASO to be orally administered and it was observed that repeated daily dosing in rats, dogs or monkeys reduces dyslipidaemia with great efficacy [161]. Another way to suppress the mRNA is to silence them using small interfering RNA (siRNA) that can be administered via lipoid nanoparticles, for which clinical trials have been successfully carried out. For example, Inclisiran (ALN-PCS) is such synthetic siRNA that specifically targets the synthesis of hepatic PCSK9 to act as a lipid-lowering therapy [162]. ALN-PCS is a synthetic siRNA drug developed by Alnylam Pharmaceuticals in the USA that can be delivered via intravenous administration into the bloodstream [163]. It inhibits the transcription of the PCSK9 gene and thus successfully reduces the plasma PCSK9 and LDL-C levels after just a single dose. Even though the first clinical trials have reported an effective clinical benefit, long term studies are yet to be conducted to assess the safety and efficacy [163]. Furthermore, adenine base editors and CRISPR adenine base editors can be used to insert a splice site mutation in the PCSK9 gene to inhibit and knockdown PCSK9 for therapeutic applications [164,165]. This has already been studied in mouse and macaque models, in order to report that this method can be fitting to treat patients with familial hypercholesterolaemia in the future [164]. PCSK9 expression can also be affected and controlled by influencing various factors involved in its production, like transcription factors. Recently it was also identified that several miRNAs can influence the PCSK9 gene expression. For example, it could be shown that miR-337-3p [166] and miR-483 [167] inhibit the transcription and translation of PCSK9 and thereby promote the uptake of LDL-C, whilst miR-552-3p enhances LDLR protein levels resulting in reduced LDL-C levels [168].

A myriad of studies have discovered synthetic and natural compounds to inhibit the gene expression of PCSK9 aside from hindering the activity of this protein. Despite the extensive research available on monoclonal antibodies to inhibit PCSK9, the end-cost makes it difficult to put this into large-scale clinical use. The development of novel approaches to silence the mRNA of PCSK9 as well as developing immunity against PCSK9 using vaccines looks extremely promising, though large clinical trials are still needed to confirm its efficacy. Further investigation on these novel therapeutic approaches might unveil a big leap in the field of cardiovascular biology.

## 7. Conclusions

PCSK9 exerts its effect on the cardiovascular system via the degradation of LDLR and by multiple mechanisms that are independent of LDLR. We have already a rather good understanding of the influence of PCSK9 on vascular biology, especially via its LDLR dependent effect, although it is clear that there are still several effects that are yet to be discovered. It has already been shown that PCSK9 inhibitors can be used to reduce cardiovascular complications in patients with well controlled LDL-C plasma levels. Therapeutically, PCSK9 antibodies inhibit the interaction between the EGF-A domain of the LDLR and PCSK9 and reduce LDL-C levels in plasma and have appeared to be used safely in clinical context. However, it remains rather unclear what the effects of PCSK9 manipulation are on LDLR-independent processes, which should therefore be an important focus point of future research. Additionally, based on the effects of the LOF mutations of PCSK9, manipulation of its gene expression might also be an additional approach to treat CVDs.

## Figures and Tables

**Figure 1 biomedicines-09-00793-f001:**
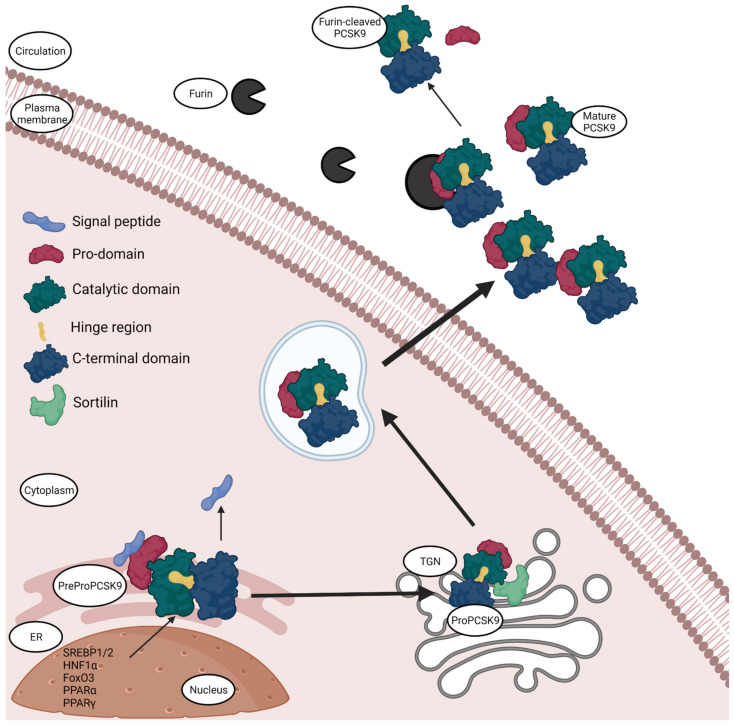
Synthesis and secretion of PCSK9. PCSK9 is synthesised in the endoplasmic reticulum (ER) with the help of transcription factors sterol-response element binding protein (SREBP-1/2), hepatocyte nuclear factor 1α (HNF1α), forkhead box O3 (FoxO3), Peroxisome proliferator-activated receptor α and γ (PPARα and PPARγ). The synthesised PCSK9 is in the form of inactive zymogen, called PreProPCSK9. PreProPCSK9 has five segments: a signal peptide, the pro-domain, catalytic domain, hinge region and the C-terminal domain; the protein then undergoes autocatalytic cleavage in the ER to lose the signal peptide and becomes ProPCSK9. Pro-PCSK9 is then transported to Trans-Golgi Network (TGN), where it interacts with Sortilin, undergoes proteolysis and the mature hetero-dimer PCSK9 is then transported in endosomes and secreted into the circulation. Normally, mature PCSK9 has its pro-domain non-covalently attached, although in circulation it can encounter furin, which then cleaves the pro-domain and releases a smaller peptide into circulation that is less active than the hetero-dimer PCSK9. The figure was created with Biorender.com (accessed on 19 May 2021).

**Figure 2 biomedicines-09-00793-f002:**
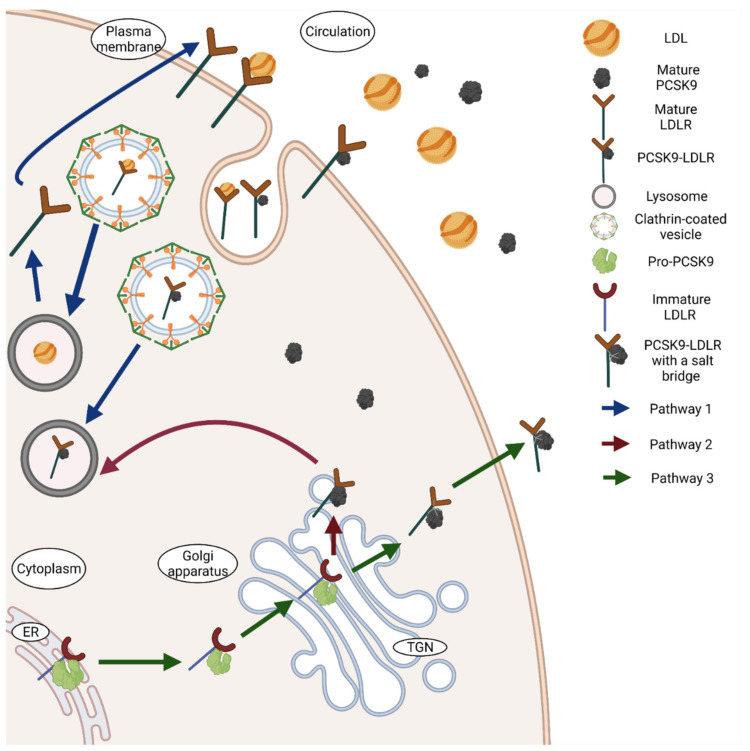
Interactions between PCSK9 and LDLR. PCSK9 interacts with LDLR through three different approaches. Pathway 1 is represented in the figure with blue arrows, while pathway 2 with red arrows and pathway 3 with green arrows. Typically, LDLR on the surface binds to the circulating LDL and transports the cholesterol particle through clathrin coated pits for degradation by lysosomes. The receptor then gets recycled back to the surface. In the presence of PCSK9, a bond between the epidermal growth factor-A (EGF-A) domain of LDLR and the catalytic domain of PCSK9 is formed. PCSK9 transports LDLR via clathrin-coated vesicles to lysosomes for degradation. In this process, PCSK9 gets degraded as well (pathway 1). Intracellularly, PCSK9 binds in the same manner with LDLR in the Trans-Golgi network (TGN) and directs the compound again towards lysosomes (pathway 2). Lastly, Pro-PCSK9 also binds to LDLR in the endoplasmic reticulum (ER) and transports the complex to Golgi apparatus, where both PCSK9 and LDLR undergo maturation. The matured PCSK9 binds to LDLR with an additional salt bridge that is released from the cells into circulation (pathway 3). The figure was created with Biorender.com (accessed on 19 May 2021).

**Figure 3 biomedicines-09-00793-f003:**
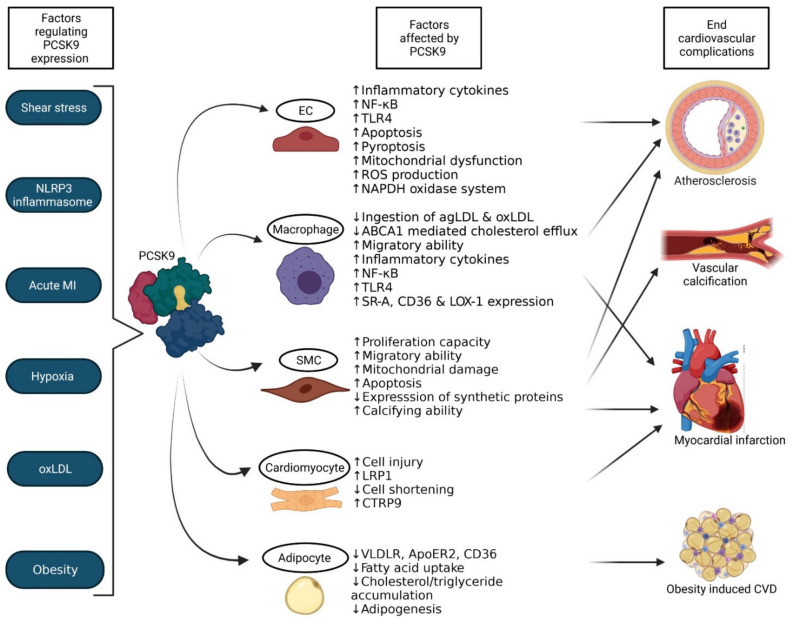
LDLR-independent mechanisms that regulate cardiovascular complications. The secretion of PCSK9 is affected by several factors, including shear stress, inflammasome, acute MI, hypoxia, oxLDL and obesity. The secreted PCSK9 then influences the expression of various factors in endothelial cells, macrophages, smooth muscle cells, cardiomyocytes and adipocytes to eventually influence the development and progression of atherosclerosis, vascular calcification, myocardial infarction and obesity induced cardiovascular disorders. ABCA1—ATP-binding cassette transporter 1; agLDL—aggregated low-density lipoprotein; ApoER2—apolipoprotein E receptor 2; CD36—Cluster of differentiation 36; CTRP9—C1q/TNF-related protein 9; CVD—cardiovascular disorder; EC—endothelial cells; LOX-1—Lectin-like oxidised low-density lipoprotein receptor-1; LRP1—low-density lipoprotein receptor-related protein 1; NADPH—Nicotinamide adenine dinucleotide phosphate; NF-κB—nuclear factor kappa-light-chain-enhancer of activated B-cells; oxLDL—oxidised low-density lipoprotein; ROS—reactive oxygen species; SMC—smooth muscle cells; SR-A—scavenger receptor-A; TLR4—Toll-like receptor 4; VLDLR—very low-density lipoprotein receptor. The figure was created with Biorender.com (accessed on 19 May 2021).

**Table 1 biomedicines-09-00793-t001:** Loss-of-function mutations.

Name of Mutation + Reference	Cause and Consequences
A443T [121,122]	Presence of a novel PCSK9 O-glycosylation site in the hinge region that promotes furin-cleavage and generates lower circulating levels of LDL-C and lower levels of fasting glucose
A522T [98] T77I [98] V114A [98] P616L [98]	Amino acid substitutions that cause hypocholesterolaemia
Ala68fsLeu82X [98]	Single nucleotide deletion in exon 1 that leads to a frameshift mutation that in turn causes the PCSK9 peptide to be shortened and not functional
C679X (rs28362286) [122,123,124,125] Y142X (rs67608943) [124]	SNPs that lead to disruption in the folding of the protein and lower concentrations of Lp(a), LDLC, oxidised phospholipid (OxPL-ApoB), fasting glucose and glycated haemoglobin
G106R [126]	GG/AG genotype in exon 2 that leads to a mutation in the prodomain due to which PCSK9 fails to undergo autocatalytic cleavage and causes an increase in the amount of surface LDLR
G236S [119] Q152D [24]	PCSK9 fails to exit the ER due to abnormal folding of the protein causing hypocholesterolaemia
N157K [126]	Causes hypocholesterolaemia, although studies do not exist on how the mutation causes the condition
N354I [119]	PCSK9 fails to undergo autocatalytic cleavage leading to the production of inactive protein
Arg46Leu [27,127]	Mutation in Exon 1 that leads to amino acid change of R46L and thereby to a lack of circulating PCSK9
Asp301Gly [27]	Mutation in Exon 6 that leads to amino acid change of D301G and thereby to a lack of circulating PCSK9
PCSK9-679X [97]	Elimination of final cysteine in the C-terminal domain that leads to PCSK9 failing to exit the ER after the protein folding is disrupted
PCSK9-FS [24]	C-terminal frameshift by which PCSK9 fails to exit the ER
Q152H (Gln152His) [24,128,129]	Amino acid substitution that prevents the autocatalytic processing of proPCSK9 inducing the reduction in circulating levels of PCSK9 and LDL-C, reduction in risk of developing CVD
R434W [103]	Alteration in the hinge region that impedes PCSK9 retention in the Trans-Golgi network that causes lower secretion levels of PCSK9
rs11206510 [130,131,132] rs11583680 (A53V) [133] rs2479409 [134] rs151193009 (R93C) [100,134]	SNPs which lead to reduced risk of CAD, peripheral artery disease, abdominal aortic aneurysm, type 2 diabetes, ischemic stroke, dementia, chronic obstructive pulmonary artery disease and cancer
rs11591147 (R46L) [126]	GT/TT genotype in exon 1 that leads to decreased levels of LDL-C and reduced risk of CVDs
S127R [118]	Mutation in pro-domain that causes low binding affinity to LDLR and increased catabolism of LDLC
S386A [126] R237W [126]	Point mutations in the catalytic domain that lead to the failure of PCSK9 to undergo autocatalytic cleavage

**Table 2 biomedicines-09-00793-t002:** Gain-of-function mutations.

Name of Mutation + Reference	Cause and Consequences
Arg499His [135] Arg496Trp [136] Asp129Gly [23]	Variations in C-terminal domain that drive the intracellular degradation of LDLR
Asp374His [23] E32K (Leu108Arg) [23] D374H [137,138]	Causes increased binding affinity to LDLR and hypercholesterolaemia
Asp374Tyr [103,136]	Mutation in the catalytic domain that improves the interaction of PCSK9 with the EGF-A domain of LDLR
Asp35Tyr [23]	Mutation that creates a novel Tyr-sulfation site to enhance the intracellular activity of PCSK9
D129G [103]	Mutation in pro-domain that leads to faster protein mobility from ER to Golgi faster in comparison to normal PCSK9
D374Y (rs137852912) [103,104,126,137,139,140] R496W (rs374603772) [104]	Causes 10–25-fold higher binding capacity to LDLR causing early CAD, atherosclerosis
D377Y [19]	Causes abdominal aortic aneurysm
Phe216Leu [99]	Decreases the circulating LDLR levels due to its destruction with the help of PCSK9 intracellularly
R215H [119] F216L [137,141] R218S [105]	SNPs that abolish furin cleavage
R357H [142]	Mutation in catalytic domain that leads to hypercholesterolaemia
R496Q [126]	Leads to hyperlipoproteinaemia
S386A [141] F216L [104,137,141]	Increases secretion of ApoB100-containing lipoproteins
S127R (rs28942111) [103,104,115,118,137,143,144]	Mutation in pro-domain that leads to increased binding affinity of PCSK9 to VLDLR and high circulating levels of VLDL, IDL and ApoB100-containing lipoproteins
Ser127Arg [136]	Variation in pro-domain that improves the chance of preventing LDLR from entering a closed conformation

## Data Availability

Not applicable.

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
