# Peer review of "PCSK9: A Multi-Faceted Protein That Is Involved in Cardiovascular Biology"

_biomedicines, 2021, doi:10.3390/biomedicines9070793_

Round 1

Reviewer 1 Report

PCSK9 is a physiological regulator of LDL receptor recycling. It is of interest since monoclonal antibodies against PCSK9 have developed as cholesterol lowering drug against statin-resistant patients. This manuscript is a well-written, comprehensive review of PCSK9 covering its biological functions, genetic polymorphisms of PCSK9, and possible drugs regulating PCSK9 expression. This review will be informative for the readers in the related fields. I would like to suggest several minor points to be considered before publication of this review.

Minor points

  1. Page 2, Figure 1. In the legend it is described “mature PCSK9 has its hetero-dimer PCSK9…”, however the illustration of PACSK 9 looks a monomer protein. One thing I am wondering is that the cleavage of the signal peptide should be proceed in ER lumen, but it looks as if it occurs the cytosolic side of TGN in the illustration. The “mature PCSK9” should be moved to the final product of PCSK9 maturation, namely the products after removing the pro-domain by furin.
  2. Section 4. It is nice to introduce various observations that PCSK9’s functions independent of LDL receptor. However, this chapter is rather descriptive. If possible, it is nice to add some mechanistic explanations howPCSK9 acts on LDLR-KO mice, EC, or SMC.
  3. Page 8, line 20. The reference cited here (ref. 4) id s review article. I think it is nice to cite an original paper reporting the overexpression of PCSK9 by SMC.
  4. Table 1. Such panel of mutations of PCSK9 are very informative. However, I wonder why N157K, and I474V are classified in loss-of-function polymorphism, since they have no information on the effects.
  5. Table 2. Why R215H, F216L, and R218S1 are classified in gain-of-function polymorphism, since they are not cleaved by furin so that mature PCSK9 will be reduced?
  6. Some of the technical words are capitalized even in the middle of sentences, while Golgi is not capitalized in some parts.
    • Line 3 of the section 2, …within the Endoplasmic reticulum and…
    • Line 25, page 5, “in the golgi or trans-golgi complex”.
    • The legend of Figure 2, “in the trans golgi network”, and “the complex to g Dose the “organisms” mead “organs”?
    • The “Berberine”, Resveratrol”, “Quercetin”, and “Liraglutide”, (page 13, line 3-11,) “Alorocumab”, and “Evolocumab” (page 14, line 4-5,), and “Inclisiran” (page 14, line 40) were capitalized.
  7. Page 11. The first 2 lines of the paragraph 2 were bold font.
  8. Page 14, line 28. The abbreviation L-IFPTA+ should be spelled out, or a short explanation may be helpful.

Reviewer 2 Report

PCSK9 is an important and is a highly studied enzyme for cardiovascular diseases. While there are some great reviews on PCSK9 inhibitors and cardiovascular diseases by Burnett and Hooper (NEJM 2018), Marc Sabatine (Nature reviews cardiology), etc. but a comprehensive review which discusses the biology of PCSK9 is not available in the literature (to my knowledge). The authors have put together this review nicely discussing loss and gain of function of PCSK9, as well as activity of PCSK9 independent of LDLR. The review provides an up-to-date knowledge on LDLR dependent and independent effects of targeting PCSK9, along with the interventions available for manipulation. 

Reviewer 3 Report

This is a nice review that collects most of the data regarding to the role of PCKS9 on cardiovascular system. I understand that while writing a manuscript authors can miss other papers that under review. By looking at the literature, I found three very recent reviews that authors should discuss what does this review add compared to them:

  • Heart Lung Circ. 2021 Jun 3;S1443-9506(21)00547-3. doi: 10.1016/j.hlc.2021.05.085.
  • Am J Pathol. 2021 May 19;S0002-9440(21)00204-2. doi: 10.1016/j.ajpath.2021.04.016.
  • Int J Mol Sci. 2021 May 30;22(11):5880. doi: 10.3390/ijms22115880.

Two nice figures have been added to the review. The readers will appreciate the effort of making such and I am sure the readers would appreciate adding new figures for the point “4. PCSK9’s activity independent of LDLR” that describes the “independent effects of PCSK9 on the cardiovascular system” which is by the way one of the aims of the review.

Results from references 13, 14 and 17 are not discussed in the review.

The following comments are related to some phrases or paragraphs on the text. Since lines are not numbered I have copied them here:

  • The authors wrote “Peroxisome proliferator-activated receptor a and g (PPARa and PPARg) regulate the gene expression of PCSK9 as well, PPARa increases the gene expression while PPARg decreases it”. How are these data related with cholesterol dependent transcription, i.e: fasting induces cholesterol synthesis and PPARa activation?. Moreover, PPARg is shown to increase triglyceride synthesis in adipocytes. If PPARg decreases PCSK9, why PCSK9 is related to adiposity and positively correlated with BMI (point 4.3 of the review).

  • The authors wrote “PCSK9 also facilitates intracellular degradation of mature LDLR present in the golgi or trans-golgi complex, before it reaches the cell surface41. In this pathway, the immature or the budding PCSK9 binds to the LDLR found in the Golgi apparatus and redirects it to the lysosomes for degradation. Unlike the extracellular degradation of LDLR, this intracellular degradation compels the involvement of catalytic activity of PCSK941. PCSK9 also interacts with LDLR in non-destructive ways. Pro-PCSK9 binds to the EGF-A repeat of the precursor of LDLR in the ER that is essential for the transport of LDLR to the Golgi apparatus wherein LDLR matures by obtaining various carbohydrates41,51. This binding is not just advantageous for the LDLR, but it also helps pro-PCSK9 to undergo auto-catalytic cleavage and thus undergo maturation”. These two functions are a bit contradictory although they are taken place in different but consecutive compartments of the secretory pathway. What do these results suggest about the PCSK9 ability to interact LDLR in RER and Golgi? Why would PCSK9 bind the EGF-A repeat that is essential for RER to Golgi transport to be redirected to the lysosomes for degradation?

  • The authors also wrote “PCSK9 inhibition reduces the accumulation of lipid particles inside monocytes and thereby also inhibits monocyte chemotaxis52. Apart from the effects on lipid uptake, PCSK9 inhibition also increases…”. Please elaborate on how PCSK9 is inhibited in references 52 and 53.

  • The authors wrote “the degradation of LDLR influences not just the cholesterol homeostasis, but also the triglyceride homeostasis, that eventually leads to…”. Could this be related to PPARalfa and PPARgamma transcriptional activity on PCKS9 gene?

  • “Studies observed that silencing of PCSK9 reduces the levels of inflammatory cytokines in aortic tissues, resulting in an attenuated plaque inflammation without changing the plasma cholesterol levels59,60.” Wouldn’t silencing of PCSK9 decrease plasma cholesterol?.
